# Concurrent remodelling of nucleolar 60S subunit precursors by the Rea1 ATPase and Spb4 RNA helicase

Valentin Mitterer[1]*[†], Matthias Thoms[2][†], Robert Buschauer[2], Otto Berninghausen[2], Ed Hurt[1]*, Roland Beckmann[2]*

[1]Biochemistry Center, University of Heidelberg, Heidelberg, Germany; [2]Gene Center, University of Munich, Munich, Germany

**Abstract** Biogenesis intermediates of nucleolar ribosomal 60S precursor particles undergo a number of structural maturation steps before they transit to the nucleoplasm and are finally exported into the cytoplasm. The AAA[+]-ATPase Rea1 participates in the nucleolar exit by releasing the Ytm1–Erb1 heterodimer from the evolving pre-60S particle. Here, we show that the DEAD-box RNA helicase Spb4 with its interacting partner Rrp17 is further integrated into this maturation event. Spb4 binds to a specific class of late nucleolar pre-60S intermediates, whose cryo-EM structure revealed how its helicase activity facilitates melting and restructuring of 25S rRNA helices H62 and H63/H63a prior to Ytm1–Erb1 release. In vitro maturation of such Spb4-enriched pre-60S particles, incubated with purified Rea1 and its associated pentameric Rix1-complex in the presence of ATP, combined with cryo-EM analysis depicted the details of the Rea1-dependent large-scale pre-ribosomal remodeling. Our structural insights unveil how the Rea1 ATPase and Spb4 helicase remodel late nucleolar pre-60S particles by rRNA restructuring and dismantling of a network of several ribosomal assembly factors.

*For correspondence:
mitterer.valentin@gmail.com
(VM);
ed.hurt@bzh.uni-heidelberg.de
(EH);
beckmann@genzentrum.lmu.
de (RB)

[†]These authors contributed
equally

**Reviewing Editor:** Ruben L
Gonzalez, Columbia University,
United States

## Editor's evaluation

This fundamental study substantially advances our understanding of the process of ribosome maturation. The authors have purified and determined the structures of several nucleolar ribosome assembly intermediates in yeast using cryo-electron microscopy (cryo-EM). The study combines genetic, biochemical, and structural analysis to provide compelling support for the conclusions the authors wish to draw. The work will be of broad interest to cell biologists, biochemists, and structural biologists.

## Introduction

Eukaryotic ribosomes consist of a small (40S) and a large (60S) subunit that are comprised of four ribosomal RNAs (rRNAs) and 79–80 ribosomal proteins. While the yeast 40S subunit contains the 18S rRNA and 33 ribosomal proteins, the 60S subunit contains the 5S, 5.8S, and 25S rRNA species and 46 ribosomal proteins. The two ribosomal subunits are assembled in a complex highly regulated and hierarchical biogenesis process that starts in the nucleolus by transcription of a common precursor RNA (the 35S pre-rRNA) and the co-transcriptional association of the first ribosomal proteins and assembly factors. Upon endonucleolytic cleavage within the internal RNA spacer element 1 (ITS1), the pre-40S and pre-60S subunits follow separate biogenesis routes in the nucleolus, nucleoplasm, and finally cytoplasm. For reviews on ribosome biogenesis see *Baßler and Hurt, 2019*; *de la Cruz et al., 2015*; *Klinge and Woolford, 2019*; *Kressler et al., 2017*.

The precursors of the large ribosomal subunit undergo a cascade of consecutive rRNA folding steps in which the initially flexible six intertwined 25S rRNA domains (named in the 5′ to 3′ direction as domains I to VI) get integrated into the developing 60S core. While in the earliest precursors the 27SA$_2$ pre-rRNA is highly flexible, the proportion of stably folded rRNA domains is strongly increased in succeeding 27SB pre-rRNA containing precursor particles (*Burlacu et al., 2017*). In line with chemical probing data, cryo-EM structures revealed that shaping of the immature rRNA occurs in a hierarchical order, in which first the 25S rRNA domains I, II, and parts of domain VI, together with the 5.8S rRNA and internal transcribed spacer 2 (ITS2), adopt a folded conformation and form the solvent-exposed back side of the pre-60S particle (structural states 1/A and 2/B) (*Kater et al., 2017*; *Sanghai et al., 2018*; *Zhou et al., 2019b*). Subsequent compaction of large parts of domains III, IV, and V results in formation of the polypeptide exit tunnel (PET) in state C to E particles. Transition of the 60S precursors to the nucleoplasm goes along with a re-positioning of the L1 stalk (helices H75-H78 of 25S rRNA domain V) allowing restructuring and incorporation of large parts of domain V and formation of the central-protuberance (CP) (state F/Nog2 particles) (*Kater et al., 2020*; *Wu et al., 2016*). As part of the CP, the 5S rRNA, which is transcribed independently from the other rRNA species and flexibly assembled at an earlier maturation stage, becomes stably integrated and structurally visible at this stage (*Leidig et al., 2014*; *Wu et al., 2016*). Upon subsequent removal of ITS2 and rotational movement of the 5S rRNA into its nearly mature orientation, the pre-60S intermediates are exported to the cytoplasm (*Gasse et al., 2015*; *Kater et al., 2020*; *Matsuo et al., 2014*; *Schuller et al., 2018*; *Thoms et al., 2015*; *Wu et al., 2016*; *Zhou et al., 2019a*). Here, final structuring of the inter-subunit side (ISS) and the peptidyl-transferase centre (PTC) completes 60S maturation and permits entrance of the assembled subunit to the pool of actively translating ribosomes (*Kargas et al., 2019*; *Ma et al., 2017*; *Malyutin et al., 2017*; *Zhou et al., 2019a*).

The accurate and efficient rRNA maturation and ribosome assembly requires ca. 80 small nucleolar RNAs (snoRNAs) and at least 200 non-ribosomal assembly factors. Among these assembly factors are a variety of energy consuming enzymes such as AAA$^+$-ATPases and RNA helicases that facilitate key RNA and protein remodelling events on nascent ribosomal intermediates (*Kressler et al., 2010*). The activity of the dynein-related AAA$^+$-ATPase Rea1 is required for two separate ATP-hydrolysis-dependent pre-60S maturation steps. In the earlier step, Rea1 releases assembly factor Ytm1 and its binding partner Erb1 from late nucleolar pre-60S particles (*Bassler et al., 2010*; *Thoms et al., 2016*; *Wegrecki et al., 2015*). In a second maturation step, which goes along with CP formation in the nucleoplasm, Rea1 dislodges assembly factor Rsa4 and thereby primes the 60S precursors for their export to the cytoplasm (*Barrio-Garcia et al., 2016*; *Kater et al., 2020*; *Matsuo et al., 2014*; *Ulbrich et al., 2009*). The interaction of Rea1 with Ytm1 and Rsa4 is established between a metal-ion-dependent adhesion site (MIDAS) at the C-terminal tip of the Rea1 tail and the ubiquitin-like (UBL) domains of its assembly factor ligands (*Ahmed et al., 2019*; *Bassler et al., 2010*; *Mickolajczyk et al., 2022*; *Ulbrich et al., 2009*). We previously revealed the architecture of Rea1 and its associated pentameric Rix1-complex, which serves as docking platform for the ATPase on nucleoplasmic pre-60S particles (*Barrio-Garcia et al., 2016*; *Kater et al., 2020*). Upon pre-60S association, a highly conserved loop motif within the Rea1 MIDAS facilitates the attachment of the C-terminal MIDAS onto the N-terminal AAA$^+$ ring domain (*Ahmed et al., 2019*; *Chen et al., 2018*; *Kater et al., 2020*; *Sosnowski et al., 2018*). This intramolecular MIDAS docking may enable a direct force-transmission upon ATP hydrolysis from the AAA$^+$ domain onto the MIDAS to dislodge the MIDAS-bound Rsa4 ligand. While there is no direct evidence that the Rix1-complex recruits Rea1 also to nucleolar 60S precursors, inhibition of Ytm1 release resulted in a shift of otherwise nucleoplasmic Rix1 particles to a nucleolar stage (*Bassler et al., 2010*). This suggests a function of the Rix1-complex for mediating Rea1 remodeling of its nucleolar substrate particles as well.

Around 20 RNA helicases involved in ribosome biogenesis have been identified so far (*Granneman et al., 2006*; *Mitterer and Pertschy, 2022*; *Rodríguez-Galán et al., 2013*). While it is suggested that these ATP-dependent enzymes promote crucial ribosomal restructuring steps including rRNA folding/re-arrangement or disassembly of snoRNAs, a precise molecular function was described only for very few RNA helicases. The DEAD-box helicase Spb4 is essential for cell viability and 60S assembly (*de la Cruz et al., 1998*; *García-Gómez et al., 2011*). Spb4 exhibits RNA-stimulated ATPase activity in vitro, which is abolished by a point mutation within the catalytic DEAD (Walker B) helicase motif (*Brüning et al., 2018*). Upon its depletion and in *spb4* catalytic mutant backgrounds, processing of the 27SB

pre-rRNA on pre-60S particles was blocked (*Brüning et al., 2018*; *de la Cruz et al., 1998*; *García-Gómez et al., 2011*). In line with this, depletion of Spb4 impaired the recruitment of the late B-factor Nog2, which is necessary for subsequent 27SB processing (*Talkish et al., 2012*). A low-resolution cryo-EM density and CRAC (UV-crosslinking and cDNA analysis) experiments suggested that Spb4 binds to pre-60S intermediates at a hinge region at the base of eukaryote-specific expansion segment 27 (ES27) within 25S rRNA domain IV (*Brüning et al., 2018*; *Kater et al., 2017*; *Sanghai et al., 2018*). However, despite the proposed pre-60S location, the function of Spb4 on these intermediates and its rRNA restructuring substrate remain unknown.

Here, we report an essential function of the RNA helicase Spb4 and its interaction partner Rrp17 for the maturation of late nucleolar pre-60S particles. Spb4 and Rrp17 bind to such intermediates directly prior to their transition to the nucleoplasm, which is mediated by the Rea1 AAA$^+$-ATPase. Consistent with a most recently published study (*Cruz et al., 2022*), our cryo-EM structure of Spb4 bound to pre-60S particles suggests how the helicase facilitates restructuring of its substrate rRNA before it can later fold into its mature-like conformation. Furthermore, the association of Spb4 with 60S precursors is required to allow the Rea1-dependent release of the Ytm1–Erb1 subcomplex. By coupling an in vitro maturation assay with isolated pre-60S particles to cryo-EM analyses, we could recapitulate this major structural transition during removal and destabilization of a large set of assembly factors, which promotes the nucleolar exit of 60S precursors.

## Results

### Spb4 and Rrp17 are enriched on late nucleolar pre-60S particles

During the transition from the nucleolus to the nucleoplasm, precursors of the 60S subunit undergo substantial changes regarding the rRNA folding status and concomitant association and dissociation of several assembly factors. Our previous cryo-EM analysis of a late nucleolar pre-60S particle (state E) purified via the dominant-lethal Ytm1 E80A mutant (this mutation impairs the Rea1-dependent release of the Ytm1–Erb1 subcomplex) revealed a low-resolution density for the DEAD-box RNA helicase Spb4 at the inter-subunit side, which was not present on earlier (states A–C) or later pre-60S maturation stages (states NE1, NE2, F) (*Kater et al., 2020*; *Kater et al., 2017*). Consistent with this observation, Spb4 was enriched on pre-60S particles isolated via Ytm1 E80A but not via wild-type Ytm1, whereas earlier nucleolar assembly factors (e.g. Noc1, Rrp5, Drs1) already dissociated at this stage (*Figure 1A*; *Kater et al., 2017*). Besides Spb4, another assembly factor, Rrp17, was accumulated on these Ytm1 E80A particles (*Figure 1A*), indicating that both, Spb4 and Rrp17, have a function during late steps of nucleolar pre-60S maturation.

As previous studies suggested a role of the *SPB4* and *RRP17* genes in ribosome biogenesis (*Brüning et al., 2018*; *de la Cruz et al., 1998*; *García-Gómez et al., 2011*; *Oeffinger et al., 2009*), we tested their essential functions by depletion of the Spb4-HA-AID and Rrp17-HA-AID proteins via an auxin-inducible degron (AID) approach, which as anticipated yielded non-viable yeast cells (*Figure 1B*, *Figure 1—figure supplement 1A*). Moreover, we analysed *spb4* point mutations of the catalytic helicase core domains (K57R: ATP-binding motif, E173A: ATP-hydrolysis motif, R360A: ATP-hydrolysis motif), and Spb4 C-terminal truncation mutants, which turned out to be lethal or exhibited a slow-growth phenotype (*Figure 1C, Figure 1—figure supplement 1B*). Subsequent affinity purifications of the helicase mutants and the C-terminal Spb4 truncations, all tagged with Flag-TEV-protein A (FTpA), showed that they still co-purify pre-ribosomal particles, however, somewhat less efficiently (*Figure 1—figure supplement 1C and D*). This indicates an essential pre-60S targeting function of the C-terminal domain (CTD), which was recently described also for DDX55, the human homologue of Spb4 (*Choudhury et al., 2021*), whereas another recent study on yeast Spb4 suggested that not the entire CTD is strictly required for pre-60S assembly (*Cruz et al., 2022*). Taken together, our results suggest that the Spb4 function on nascent 60S pre-ribosomes depends on both the catalytic and C-terminal domain.

In contrast to the tested *spb4* mutants, affinity purifications via wild-type Spb4 co-enriched a pattern of assembly factors highly similar to Ytm1 E80A particles (state E) and contained large amounts of Rrp17 (*Figure 1D*, left panel). Vice versa, Rrp17-FTpA affinity purifications co-enriched pre-60S particles with a comparable assembly factor composition and Spb4 but in contrast to Spb4 purifications also a few downstream nucleoplasmic assembly factors like Nog2, Rsa4, and the ITS2 foot-factor

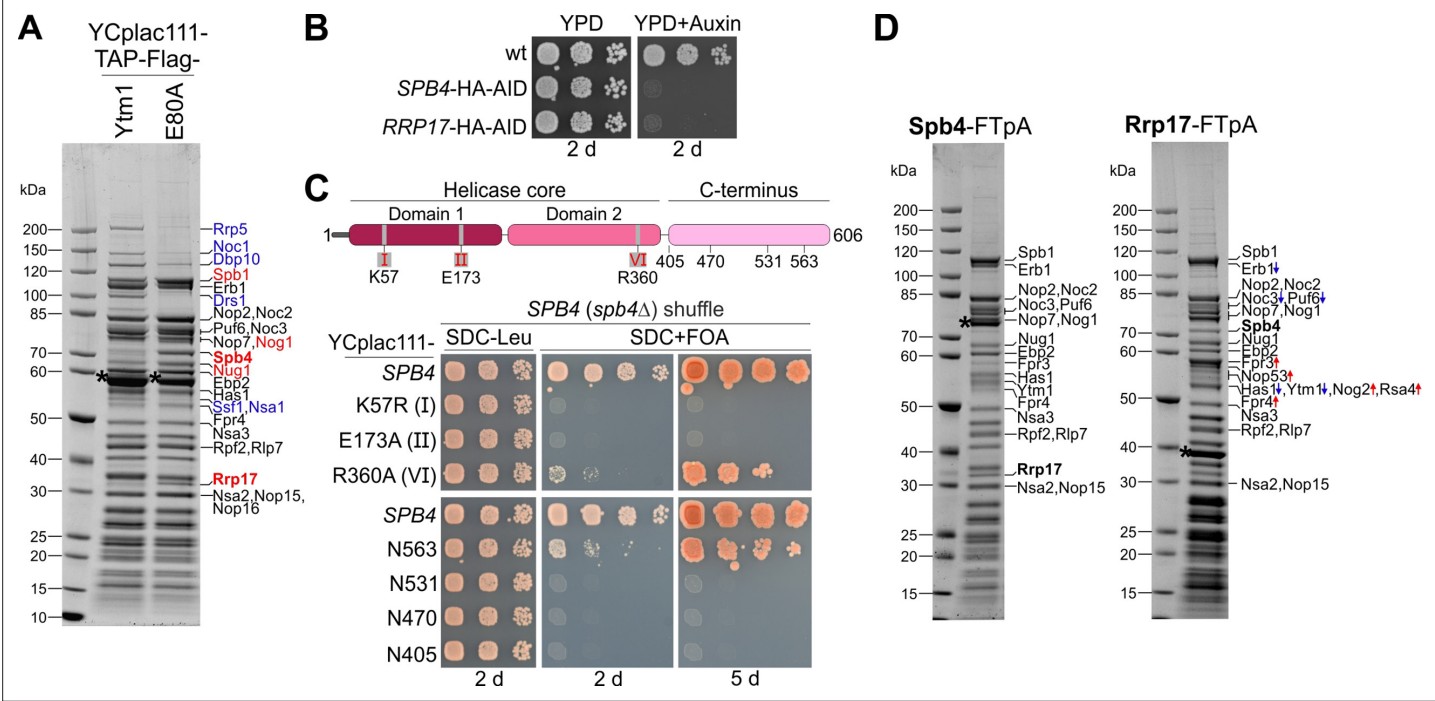

**Figure 1.** Spb4 and Rrp17 bind to late nucleolar 60S precursors. (**A**) Spb4 and Rrp17 are enriched on Ytm1 E80A particles. Plasmid-borne TAP-Flag tagged wild-type *YTM1* or the *ytm1* E80A mutant were affinity purified and final eluates analysed by SDS-PAGE and Coomassie staining. (**B**) *SPB4*-HA-AID and *RRP17*-HA-AID strains were spotted in tenfold serial dilutions on YPD plates without (YPD) or containing auxin (YPD + Auxin) and incubated for 2 d at 30°C. (**C**) Upper panel: Spb4 protein domain organization. Amino acids within the conserved helicase core signature motifs I (K57: ATP-binding), II (E173: ATP hydrolysis), and VI (R360: ATP-hydrolysis) that were mutated in this study are indicated. Lower panel: A *SPB4* (*spb4Δ*) shuffle strain was transformed with plasmids harbouring wild-type *SPB4* or indicated *spb4* mutant alleles. Transformants were spotted in tenfold serial dilutions on SDC-Leu or 5-FOA containing plates (SDC + FOA) and growth was monitored after incubation at 30°C for the indicated days. (**D**) Spb4-FTpA (left panel) and Rrp17-FTpA (right panel) affinity purifications enrich late nucleolar pre-60S particles. Pre-60S particles were isolated via chromosomally encoded Spb4- or Rrp17-FTpA. Final eluates were analysed by SDS-PAGE and Coomassie staining and indicated protein bands identified by mass spectrometry. Bands that show an increased or decreased relative intensity in the Rrp17 eluate compared to the Spb4 sample are depicted in red and blue, respectively. Note that the nucleoplasmic factors Nog2 and Rsa4 were detectable only in the Rrp17 sample. Bait proteins are marked with an asterisk.

The online version of this article includes the following source data and figure supplement(s) for figure 1:

**Source data 1.** Unedited and uncropped Coomassie-stained gels shown in *Figure 1*.

**Figure supplement 1.** Affinity purifications using Spb4 mutants as bait protein.

**Figure supplement 1—source data 1.** Unedited and uncropped Coomassie-stained gels and western blots shown in *Figure 1—figure supplement 1*.

Nop53 (*Figure 1D*, right panel). Based on these affinity purifications, we conclude that both, Spb4 and Rrp17, join the maturation pathway around nucleolar state E. While Spb4 subsequently dissociates during or shortly after release of Ytm1–Erb1 by Rea1, Rrp17 may remain associated on downstream intermediates upon Nop53 association (Nop53 binds pre-60S particles at state NE1 upon Ytm1–Erb1 removal).

## Depletion of Spb4 and Rrp17 arrests pre-60S maturation at a stage directly before nucleolar exit

To elucidate which particular pre-60S maturation step(s) require Spb4 and Rrp17, we assessed the impact of their cellular depletion on the assembly factor composition of purified pre-60S intermediates. Therefore, we generated chromosomal AID fusions of *SPB4* or *RRP17* in strains expressing FTpA-tagged Nop7, Nug1, or Arx1 and isolated these assembly factors from cells in presence of Spb4-HA-AID and Rrp17-HA-AID (-auxin) or after their auxin-induced degradation (+auxin) (*Figure 2*). In the absence of Spb4 or Rrp17, pre-60S maturation was effectively arrested at a late nucleolar stage, as apparent by the accumulation of assembly factors Ytm1–Erb1, Spb1, and Has1 on Nug1- and Nop7-derived particles, whereas the recruitment of Nop53 and downstream nucleoplasmic assembly factors

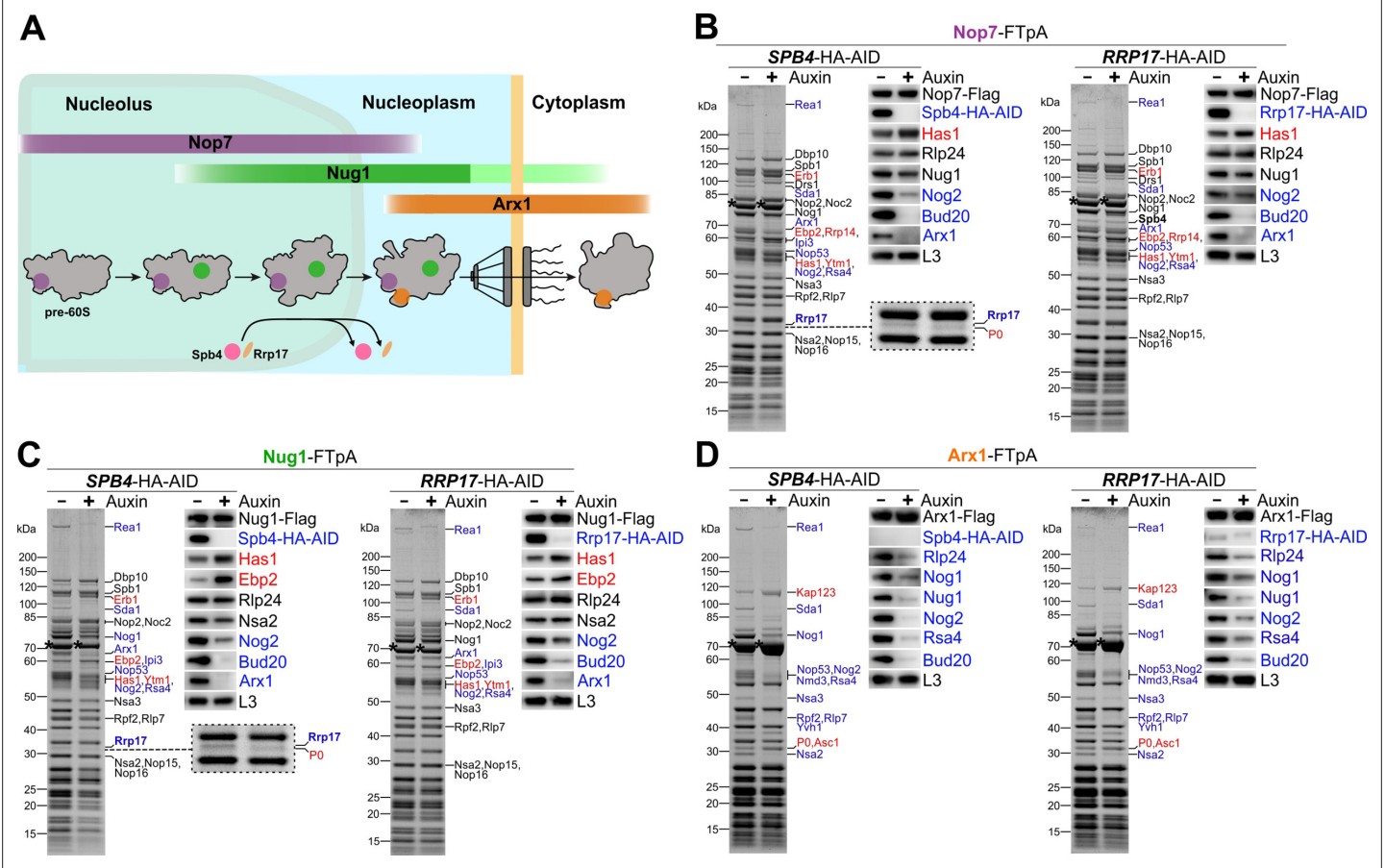

**Figure 2.** Spb4 and Rrp17 depletion arrests pre-60S maturation at a late nucleolar stage. (**A**) Colour-coded bait proteins selected for affinity purifications are associated with distinct maturation intermediates along the 60S biogenesis pathway in the nucleolus, nucleoplasm, and cytoplasm: Nop7 (violet), Nug1 (green), and Arx1 (orange). The proposed assembly/disassembly stage of Spb4 and Rrp17 is indicated. (**B–D**) Nop7-FTpA, Nug1-FTpA, and Arx1-FTpA were affinity purified from cells in the presence of Spb4-HA-AID or Rrp17-HA-AID (-Auxin) or upon their auxin-induced proteasomal degradation (+Auxin). Final eluates were analysed by SDS-PAGE and Coomassie staining (left panels) or western blotting using indicated antibodies (right panels). Depicted Coomassie-stained protein bands were identified by mass spectrometry. Bait proteins are marked with an asterisk. Protein bands with increased or decreased intensity in the depleted samples are indicated in red and blue, respectively. For a better visualization, the area of the Coomassie-stained gels of the Nop7 and Nug1 eluates to which Rrp17 migrates is enlarged (**B** and **C**, left panels; note that the additional protein band appearing in the Spb4-HA-AID-depleted samples was identified as ribosomal protein P0).

The online version of this article includes the following source data for figure 2:

**Source data 1.** Unedited and uncropped Coomassie-stained gels and western blots shown in *Figure 2B*.

**Source data 2.** Unedited and uncropped Coomassie-stained gels and western blots shown in *Figure 2C*.

**Source data 3.** Unedited and uncropped Coomassie-stained gels and western blots shown in *Figure 2D*.

such as Nog2 and Bud20 to these stalled maturation intermediates was impaired (*Figure 2B and C*). This suggests an important function of both Spb4 and Rrp17 for pre-60S maturation before Ytm1– Erb1 dissociation and transition of pre-60S particles to the nucleoplasm. Consistently, the export factor Arx1 that joins at a nucleoplasmic stage failed to efficiently co-purify pre-ribosomes upon Spb4 or Rrp17 depletion and was instead present mainly in a free form or associated with mature ribosomes or non-ribosomal factors (e.g. importins) (*Figure 2D*). Notably, while the association of Spb4 with Nug1- and Nop7-derived particles was a prerequisite for recruitment of Rrp17, the presence of Rrp17 was apparently not strictly required for Spb4-recruitment to these 60S precursors (*Figure 2B*, right panel; note that in *Figure 2C* Spb4 is co-migrating with the Nug1-Flag bait protein as revealed by mass spectrometry analyses).

Taken together, our data revealed an essential function of Spb4 and Rrp17 during final steps of nucleolar pre-60S maturation directly prior to release of the Erb1–Ytm1 subcomplex and subsequent

assembly of Nop53. Thereby, Spb4 may associate with 60S precursors shortly before Rrp17 and help to mediate its incorporation.

## Cryo-EM structure of the Spb4 particle

To obtain insight into the molecular function of Spb4, we sought to analyse the structure of the helicase bound to its pre-ribosomal 60S substrate particle and performed single-particle cryo-EM analysis with affinity-purified Spb4-FTpA intermediates. The major class of particles (40.85% of particles after 2D classification), which closely resembles the structural state E (*Kater et al., 2017*), was resolved at an overall resolution of 2.7 Å (*Figure 3A and B*, *Figure 3—figure supplement 1A–C*, and *Figure 3—figure supplement 2A–E*), whereas a minor class corresponded to state D at an overall resolution of 3.0 Å (*Figure 3—figure supplement 1A–C* and *Figure 3—figure supplement 2F–I*). The latter class contains in addition the Nsa1 module (consisting of assembly factors Nsa1, Rpf1, Mak16, and Rrp1) at the solvent-exposed pre-60S side but is otherwise identical to the better resolved state E (*Figure 3—figure supplement 3A*). The higher resolution of the state E compared to our previous study (*Kater et al., 2017*) allowed us to identify and build molecular models of assembly factors Spb4, Rrp17, Noc2, and Loc1, as well to resolve additional parts of Noc3, Spb1, Nsa3, and of the ITS2 RNA (*Figure 3A and B*, *Figure 3—figure supplement 2A–E*). At the inter-subunit side Spb4 is in contact with Rrp17 and located on top of the immature rRNA helices H62 and H63 in direct vicinity to the Ytm1–Erb1 complex (*Figure 3A, B, H and I*). The intertwined Loc1 is located at the solvent-exposed pre-60S side where it engages in multiple protein interactions with assembly factors Noc3, Spb1, Nip7, Erb1, Ebp2, Brx1, and ribosomal protein Rpl15 (eL15) (*Figure 3A and B*, *Figure 3—figure supplement 3B*). Next to Loc1, Noc3 interacts with the C-terminal part of its heat-repeat domain with its binding partner Noc2 (*Figure 3B*, *Figure 3—figure supplement 3C*). While the middle part of Noc2 (aa 86–176) is not resolved, the C-terminal Noc2 heat-repeat domain and the N-terminal part of the protein are visible, and both contribute to establish the interaction with Noc3. Besides its interaction with Noc3, Noc2 is anchored to the particle by its elongated N-terminus that protrudes deeply into the 60S core. There, the Noc2 N-terminus contacts the immature H35 rRNA, which likely requires Noc2 dissociation to adopt its mature conformation (*Figure 3—figure supplement 3C*). We were also able to resolve additional parts of the ITS2 spacer RNA (nucleotides 60–65 and 213–226) and the C-terminus of Nsa3 (aa 324–353) at the prominent pre-60S 'foot' structure (*Figure 3—figure supplement 3D*). This unveiled internal base-pairing of ITS2 bases 62–65 with 213–216, which is in line with a proposed 'ring-pin' model for the ITS2 structure based on chemical probing data (*Burlacu et al., 2017*). Furthermore, it revealed how the ITS2 RNA and Nsa3 are twined around and enclose assembly factor Nop15 (*Figure 3—figure supplement 3D*), which therefore can presumably dissociate only upon ITS2 processing and subsequent release of the foot factors (*Gasse et al., 2015*; *Schuller et al., 2018*).

## The α-helical Rrp17 connects Spb4 and the methyltransferase Spb1

While the methyltransferase Spb1 was already visualized in previous cryo-EM structures (*Kater et al., 2020*; *Kater et al., 2017*), we were able to model additional parts (between aa 347–570) of the so far unresolved widely meandering central region of the protein revealing interactions with Noc3, Rpl25 (uL23), Rpl30 (eL30), and Rpl34 (eL34) (*Figure 3A and B*, *Figure 3—figure supplement 4A*). Notably, we identified a ligand within the binding pocket of Spb1's methyltransferase domain (MTD), which is most probably S-adenosyl homocysteine (SAH) (*Figure 3E*, *Figure 3—figure supplement 4B*), suggesting that methylation of its 2'-O-ribose target in nucleotide G2922 of the A-site loop (*Lapeyre and Purushothaman, 2004*) has occurred at this stage. The G2922 methylation was most recently shown to be monitored by the GTPase Nog2 establishing a crucial checkpoint for pre-60S maturation (*Yelland et al., 2023*). To facilitate this modification, the very N-terminal part of Spb1 (aa 4–27) may play an important role for target recognition as it clamps 25S rRNA H92 that harbours nucleotide G2922 and thereby suitably positions it for methylation (*Figure 3E and F*). In agreement with 2'-O-ribose methylation of G2922 at the analysed stage, we observed Gm2922 in the corresponding RNA density as well as adjacent Um2921 (*Figure 3E*, *Figure 3—figure supplement 4B*), which is in a redundant function methylated by Spb1 or the snR52 C/D box snoRNP at an earlier pre-60S biogenesis step (*Bonnerot et al., 2003*; *Lapeyre and Purushothaman, 2004*).

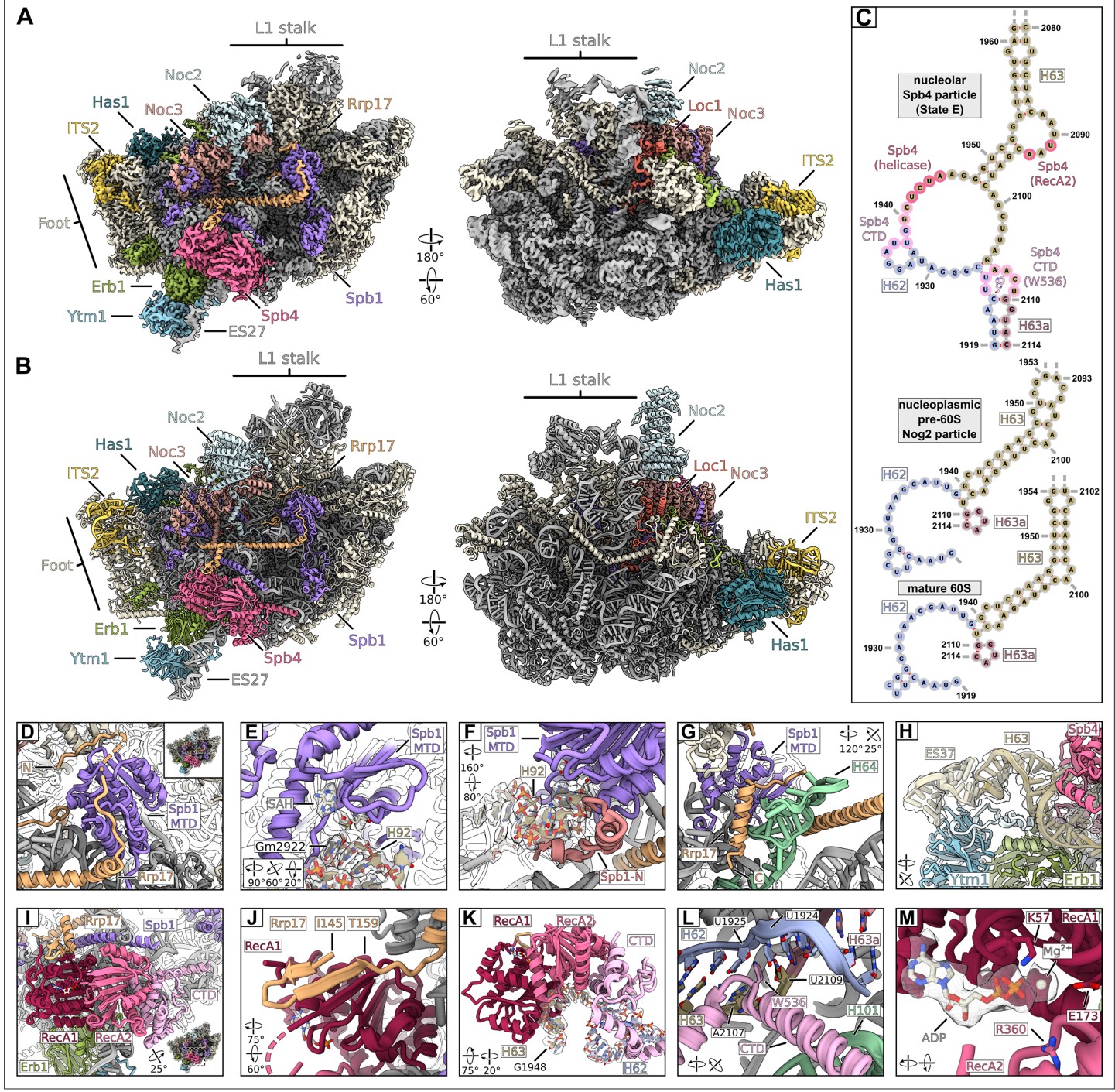

**Figure 3.** Cryo-EM structure of the nucleolar state E pre-60S subunit. (**A, B**) Cryo-EM density and model in two orientations, ribosome biogenesis factors relevant to the study are coloured and labelled. Ribosomal proteins and rRNA are shown in grey and additional biogenesis factors in beige. (**C**) Cryo-EM model-based secondary structure of H62, H63, and H63a of the rRNA in the nucleolar state E, the nucleoplasmic Nog2 particle (PDB-3jct) and mature ribosomes (PDB-6tb3) as comparison. Nucleotides interacting with Spb4 and the intercalating Trp536 of Spb4 are indicated. (**D–M**) Magnified regions focussing on Spb1, Spb4, and Rrp17. (**D**) The Spb1-MTD interacting with the N-terminus of Rrp17. (**E, F**) Spb1-MTD interacting with the A loop H92. The positioned Gm2922 and S-adenosyl homocysteine (SAH) are labelled and the segmented densities for H92 and SAH are shown. (**F**) The N-terminus (highlighted in red) of the Spb1-MTD clamps around H92 and helps positioning G2922. (**G**) Interaction of the Rrp17 C-terminus with H64 and the Spb1-MTD. (**H**) Ytm1 and Erb1 in contact with H63 and ES27. Insertions of Ytm1 (between ß16-ß17) and Erb1 (between ß8-ß9) are coloured in light blue and light green, respectively. The segmented density for ES27 and H63 is shown. (**I**) Overview of Spb4 bound to the state E pre-60S subunit. The RecA domains (RecA1, RecA2) and the C-terminal domain (CTD) are coloured and labelled. (**J–L**) Interaction of Spb4 with Rrp17

*Figure 3 continued on next page*

*Figure 3 continued*

(**J**) and the immature H62/H63/H63a rRNA (**K, L**). (**M**) The Spb4 helicase core in complex with ADP. Residues mutated in this study are labelled and the segmented density for ADP is shown.

The online version of this article includes the following figure supplement(s) for figure 3:

**Figure supplement 1.** Cryo-electron microscopy data processing scheme.

**Figure supplement 2.** Local resolution distribution and Fourier shell correlation (FSC) curves for state E, state D, and state E lacking Rrp17.

**Figure supplement 3.** Structural details of nucleolar pre-60S states D/E.

**Figure supplement 4.** Structural details of Spb1 and Spb4.

**Figure supplement 5.** Importance of the Spb4–Rrp17 β-sheet augmentation for cell growth and protein–protein interaction.

**Figure supplement 6.** Cryo-EM structure of the state preceding state E lacking Rrp17 and the Spb1-MTD.

The α-helical Rrp17 is present at the inter-subunit side clamped between the MTD of Spb1 and the helicase core of Spb4 (*Figure 3A and B*, *Figure 3—figure supplement 5A*). While the essential unstructured Rrp17 N-terminus is meandering across the MTD of Spb1 (*Figure 3D*), a more central part of the protein (aa 145–159) is reaching towards Spb4 forming an anti-parallel β-sheet with the N-terminal RecA-like domain of the helicase (*Figure 3I and J*, *Figure 3—figure supplement 5A*). Prompted by this β-sheet augmentation, we analysed the growth of yeast cells upon mutation of the corresponding Rrp17 amino acids (154–160). As observed for N-terminal (Spb1 interaction) but not C-terminal Rrp17 truncation mutants, disruption of the β-sheet augmentation in *rrp17* 154–160>A and 154–160>R mutant cells yielded slow-growth phenotypes, revealing the importance of this Rrp17–Spb4 contact (*Figure 3—figure supplement 5B*). Moreover, a yeast two-hybrid interaction between Spb4 and Rrp17 was observed, mediated by the central part of Rrp17 (aa 106–177) that includes amino acids 154–160 (*Figure 3—figure supplement 5C*). The C-terminal part of Rrp17 (aa 209–235) is reaching back and contacts again the Spb1 MTD in proximity of its N-terminus, as well as 25S rRNA H64 (*Figure 3G*).

## Spb4-containing pre-60S intermediates up- and downstream of state E

In the major class of the obtained state E-like Spb4 particles, Rrp17 and Spb1 including its N-terminal MTD were already incorporated as described above. In addition, the Spb4 purifications contained a mixture of distinct NE1-like states downstream of Ytm1–Erb1 removal (*Kater et al., 2020*) as well as state E-like particles with only weak densities for the L1 stalk, Rrp17, and the Spb1 MTD, potentially representing intermediates upstream of state E (*Figure 3—figure supplement 1C*). Moreover, analysis of another independent dataset of purified Spb4 particles, which underwent an only partially efficient maturation reaction, revealed an additional E-like state (resolved at 3.1 Å) lacking the corresponding Rrp17 and Spb1 MTD densities but show stable incorporation of the L1 stalk (*Figure 3—figure supplement 1D–F*, *Figure 3—figure supplement 2J–M*, *Figure 3—figure supplement 6*). We conclude that this state may indeed represent a novel maturation intermediate directly preceding state E. Taken together, our cryo-EM analyses suggest that Spb4 is assembled around state D/E shortly prior to final Spb1 MTD and Rrp17 incorporation, which agrees with the biochemical protein data of purified pre-60S particles upon Spb4 and Rrp17 depletion (*Figure 2*).

## Spb4 accommodates ADP and promotes H62/H63/H63a restructuring

Our cryo-EM structure gave several insights into the molecular interactions of the Spb4 catalytic and C-terminal domains on the pre-60S particle (*Figure 3A–C , and H–L*). Most importantly, we observe a single-stranded rRNA (nucleotides U1942 to A1945 of 25S rRNA H63) within the helicase core active site (*Figure 3K*, *Figure 3—figure supplement 4C*). The accommodation of the rRNA substrate between the two RecA-like domains induces bending and strand separation of the rRNA around the base of ES27, resulting in an alternate base-pairing of helices H62/H63/H63a compared to nucleoplasmic maturation intermediates and mature 60S subunits (*Figure 3C and K*). This may explain why the rRNA area at the base of 25S domain IV initially appears to form stable duplexes, while it becomes more flexible and accessible for chemical modification in presence of Spb4, suggesting that the helicase disrupts this region upon its association (*Brüning et al., 2018*; *Burlacu et al., 2017*). In addition to the catalytic domain, Spb4's essential CTD appears significantly involved in inducing substrate RNA strand disruption and establishing this alternate conformation. In the obtained substrate-bound state,

the first half of the CTD (aa 406–499) is tightly docked onto the C-terminal RecA-like domain (RecA2) and binds H62/H63 nucleotides A1936 to C1941, thereby maintaining separation of the rRNA strands (*Figure 3C, K and L*, *Figure 3—figure supplement 4C*). Furthermore, a conserved tryptophan (W536) within the flexible C-terminal tail of the CTD (aa 500–606) intercalates between nucleotides of the immature H62/H63/H63a rRNA, which later adopts its mature-like fold in nucleoplasmic pre-60S particles (*Figure 3C and L*).

Two main conformations for the catalytic RNA helicase core domains have been described, which depend on the nucleotide and substrate binding state. Cooperative ATP and RNA substrate binding bring the two RecA domains in a closed conformation, while subsequent ATP hydrolysis usually leads to substrate release and re-opening of the RecA domains (*Andersen et al., 2006*; *Bono et al., 2006*; *Mallam et al., 2012*; *Sengoku et al., 2006*; *Theissen et al., 2008*). Examination of the catalytic Spb4 core in our structure revealed that, while the two RecA domains are in a clear closed conformation and tightly bound to the substrate RNA, the helicase surprisingly accommodates in its binding pocket a nucleotide which we could unambiguously identify as ADP (*Figure 3M*, *Figure 3—figure supplement 4D and E*). Thus, Spb4 has already hydrolysed ATP at a previous step but remains associated with pre-ribosomes in a post-catalytic and still ADP-bound closed state and preserves its re-arranged rRNA substrate in the observed immature alternate conformation as also observed in a recent structure (*Cruz et al., 2022*). This unusual behaviour can be explained by the requirement of ATP binding and hydrolysis to provide the energy for binding and local strand melting of the rRNA which is then followed by a subsequent stabilization of this state through the observed interactions of Sbp4 with several assembly factors such as Rrp17. As suggested by *Cruz et al., 2022*, it would also allow for a concerted release of Spb4 together with Ytm1 and other factors upon Rea1 activity, thereby facilitating the timely re-annealing of the H62-H63 rRNA module in the mature conformation as observed in the subsequent NE states. Thus, the interaction and activity of Sbp4 appears to set the stage for the correct formation of the H61-H62-H63-H64 rRNA region as an important prerequisite for the further maturation of rRNA domain IV.

## Rea1-dependent nucleolar pre-60S maturation can be induced on isolated Spb4 particles in vitro

Next, we wanted to analyse in further detail how the Rea1 ATPase extracts Ytm1–Erb1 from nucleolar pre-60S particles. Therefore, we aimed to induce Rea1-dependent maturation of isolated TAP-Ytm1 pre-60S intermediates in vitro. We incubated Ytm1-derived particles with affinity purified Rea1, the pentameric Rix-complex (i.e. Rix1$_2$–Ipi3$_2$–Ipi1; this complex is suggested to guide Rea1 to its pre-ribosomal substrates), and ATP. However, upon subsequent re-isolation of pre-60S particles via Rpl3-Flag, the Ytm1–Erb1 complex remained fully associated with the pre-ribosomes (*Figure 4—figure supplement 1A*). Strikingly, performing this maturation assay with Spb4-purified pre-60S particles exhibited an efficient release of the Ytm1–Erb1 complex and, remarkably, the DEAD-box helicase Has1 (*Figure 4A*, lane 8). The release of these assembly factors was more efficient in presence of the Rix1-complex, however, the addition of Rea1 and ATP was in principle sufficient to perform the restructuring step, at least in this in vitro setup with excess of purified Rea1 (*Figure 4A*, lane 7). The in vitro maturation also occurred when the slow growing catalytic *spb4* R360A mutant was used (*Figure 1C*, *Figure 4—figure supplement 1B and D*), which indicates that ATP-hydrolysis by Spb4 is not required for this maturation step. However, since in vitro maturation was inducible on Spb4- but not on Ytm1-derived pre-60S particles that co-purify only small amounts of the helicase Spb4 (*Figure 1A*), we conclude that pre-60S recruitment of Spb4 and potentially also Rrp17 is a prerequisite for this remodelling step. Indeed, also incubation of Rrp17 particles with Rea1, Rix1-complex, and ATP resulted in limited but clearly detectable release of the sub-stoichiometrically associated Ytm1–Erb1 complex (*Figure 4—figure supplement 1C*).

## Ytm1–Erb1 release promotes the dissociation of the RNA helicase Has1

While Rea1's direct substrate Ytm1 is recruited to the pre-60S subunit via tight interactions with the C-terminal β-propeller domain of Erb1 near the bottom of the foot-structure (*Kater et al., 2017*; *Thoms et al., 2016*; *Wegrecki et al., 2015*), the RNA helicase Has1 is bound at a distant site over H16 at the top of the foot and is contacted by the meandering unstructured Erb1 N-terminal tail (*Figure 4—figure supplement 2A*; *Kater et al., 2017*; *Zhou et al., 2019b*). The nucleotide binding

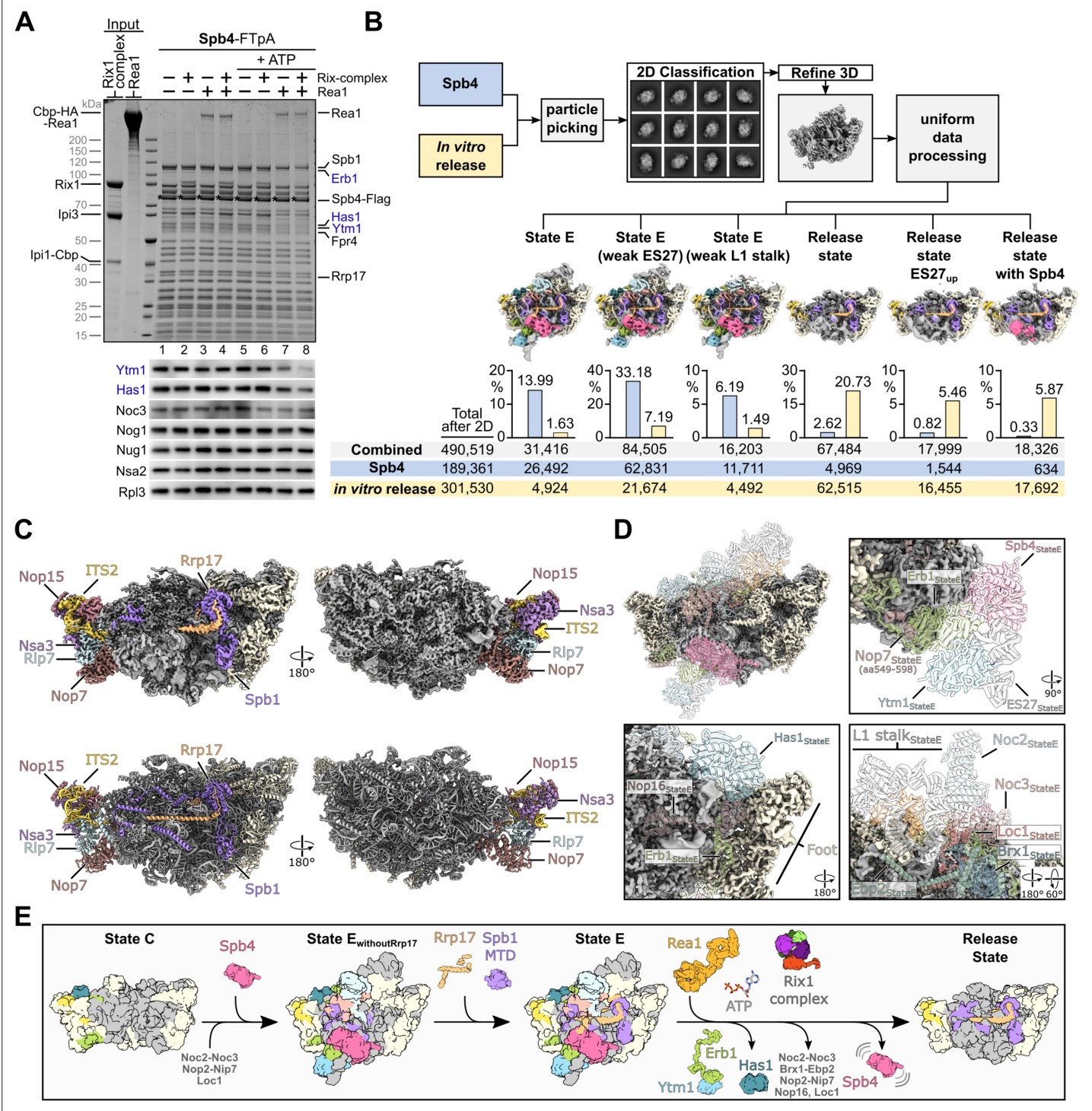

**Figure 4.** Rea1-dependent remodelling induced on isolated Spb4 particles in vitro. (**A**) Rea1 releases the Ytm1–Erb1 subcomplex together with RNA helicase Has1 from pre-60S particles. TEV-eluates of pre-60S particles isolated via Spb4-FTpA were as indicated incubated with purified Rea1, Rix1-complex (Rix1$_2$–Ipi3$_2$–Ipi1) and/or 3 mM ATP for 45 min at room temperature. Still bound material was re-isolated via Spb4-Flag by incubation with anti-Flag agarose beads and eluted with Flag peptide. Final eluates were analysed by SDS-PAGE and Coomassie staining (upper panel) or western blotting using the indicated antibodies (lower panel). The Spb4-Flag bait protein is marked with an asterisk and the released assembly factors Ytm1, Erb1, and Has1 are indicated in blue (**B**) Single-particle cryo-EM analyses of Spb4 control and in vitro matured pre-60S samples. Micrographs of the untreated Spb4 control and the in vitro release samples (corresponding to lanes 1 and 8 in (**A**), respectively) were combined for data processing. After reference free particle picking and 2D classification, the combined dataset was processed and six final classes are shown (maps are 3 x binned and coloured according to the colour scheme in *Figure 3*). The diagrams indicate the percentage of particles from the Spb4 and in vitro release dataset

*Figure 4 continued on next page*

*Figure 4 continued*

corresponding to the individual cryo-EM reconstruction. Combined particle numbers of each class as well as the particle numbers of the individual datasets (Spb4 control and in vitro release) are shown underneath. (**C**) Cryo-EM map (upper panels) and model (lower panel) of the main release state. Spb1, Rrp17 and the foot factors are coloured and labelled and additional biogenesis factors are highlighted in beige. (**D**) Overview and magnified regions of the release state cryo-EM volume with ribosomal proteins and rRNA coloured in grey and biogenesis factors and ITS2 RNA coloured in beige. Superposition of transparent molecular models from the preceding state E indicating factors and rRNA which are destabilised/released in the release state class. (**E**) Model summarising the results of this study. Spb4 is assembled at a late nucleolar stage and remains attached in a helicase-core closed conformation upon ATP hydrolysis. Rrp17 is incorporated at state E, at which also the Spb1-MTD gets positioned for methylation of G2922. Rea1-dependent release of Ytm1–Erb1 facilitates large-scale remodelling of the particle and transition to the NE1-like release state.

The online version of this article includes the following source data and figure supplement(s) for figure 4:

**Source data 1.** Unedited and uncropped Coomassie-stained gels and western blots shown in *Figure 4A*.

**Figure supplement 1.** In vitro maturation assays utilizing distinct substrate particles.

**Figure supplement 1—source data 1.** Unedited and uncropped Coomassie-stained gels and western blots shown in *Figure 4—figure supplement 1*.

**Figure supplement 2.** Catalytically inactive Has1 K92A protein is dislodged from Spb4 particles upon Ytm1–Erb1 release.

**Figure supplement 2—source data 1.** Unedited and uncropped Coomassie-stained gels and western blots shown in *Figure 4—figure supplement 2*.

**Figure supplement 3.** Comparison of Spb4, in vitro release, and control samples.

**Figure supplement 4.** Sorting scheme of the combined Spb4 and in vitro release datasets.

**Figure supplement 5.** Local resolution and Fourier shell correlation (FSC) curves of the observed release states.

**Figure supplement 6.** Cryo-EM maps of the release state with ES27up and the release state containing Spb4.

pocket of Has1 was found empty in our and a previously published pre-60S cryo-EM structure (*Zhou et al., 2019b*). To address whether upon addition of ATP the catalytic activity of Has1 might be activated during the observed pre-60S release, we performed the in vitro maturation assay with Spb4 particles derived from *HAS1*-HA-AID-depleted cells in presence of a catalytically inactive, non-viable *has1* K92A mutant impaired in nucleotide-binding (*Figure 4—figure supplements 1D and 2B*, left panel; *Rocak et al., 2005*). Upon incubation with Rea1, the Rix1-complex, and ATP, wild-type and mutant Has1 proteins were released with similar efficiency together with the Ytm1–Erb1 complex, demonstrating that the removal of all three factors occurs independently from Has1's catalytic activity (*Figure 4—figure supplement 2B*, left panel; compare lanes 4 and 8). Furthermore, the Ytm1–Erb1 release could be induced also from particles in absence of any Has1 protein (*Figure 4—figure supplement 2B*, right panel; lane 12). Thus, the Rea1-dependent pre-60S release of the Ytm1–Erb1 complex occurs before and independently of Has1, and in turn results in displacement of the helicase, which is potentially pulled off by its interaction with the intertwined Erb1 tail.

## Cryo-EM analysis of in vitro matured Spb4-derived pre-60S ribosomal particles

To follow the conformational re-arrangements induced by Rea1/Rix1-complex on Spb4-derived pre-60S particles, we analysed the in vitro matured particles by single-particle cryo-EM. Initial classification of the untreated Spb4 sample, the in vitro control sample (Spb4 particles, Rea1/Rix1-complex, no ATP), and the in vitro release sample (Spb4 particles, Rea1/Rix1-complex, plus ATP) indicated a significant formation of in vitro matured pre-60S particles (44.1% of all particles), resembling structural state NE1 particles (*Kater et al., 2020*; *Figure 4—figure supplement 3*). In contrast, only 7.4 and 7.0% of such NE1-like particles, respectively, were seen in the untreated Spb4 and Spb4 in vitro control samples. To better compare the ratios and classes within the untreated Spb4 control and the in vitro release samples, and to avoid any sorting bias of the individual datasets, we decided to join the micrographs of both samples and perform uniform data processing (*Figure 4—figure supplements 4 and 5*). After extensive sorting, we identified six major classes, three state E-like states containing Ytm1–Erb1 and three released states similar to the described NE1 state (*Figure 4B*, *Figure 4—figure supplements 4–6*). The Ytm1–Erb1-containing state E-like states were mainly present in the untreated Spb4-FTpA sample (53.36%; only 10.31% in the in vitro release sample), whereas the release states strongly increased in the case of the in vitro release sample (32.06%; only 3.77% in the Spb4 control sample). We conclude that Rea1/Rix1-complex/ATP-induced remodelling of Spb4 particles in vitro resulted in maturation from E-like to distinct NE1-like states. All three release states have strong

resemblance to the previously described NE1 state (obtained from Nop53 affinity purifications) but lack the foot-factor Nop53. However, we also find a minor particle class containing Nop53, which indicates that the attachment of Spb4 and Nop53 to pre-60S intermediates might not be strictly mutually exclusive (*Figure 4—figure supplement 4*).

Notably, all obtained release state particles still contain Rrp17 bound to the Spb1 MTD (*Figure 4B and C*), which is in line with Rrp17-FTpA purifications suggesting that the protein is still present on intermediates downstream of Spb4 (*Figure 1D*). In contrast, the Ytm1–Erb1 complex, the helicase Has1 and also Spb4 were no longer visible in the in vitro released states. Since Spb4 was our bait for specimen purification, it appears that Spb4 remained attached to the in vitro matured pre-60S particles in a flexible way and hence was not detectable by cryo-EM. Indeed, one of the released states showed a fuzzy density in the area where the Spb4 helicase core is located in the preceding state E, and rigid body fitting of the Spb4 helicase core clearly suggests that the poorly resolved density corresponds to Spb4 (*Figure 4—figure supplement 6B*). Moreover, the densities for several other assembly factors such as Loc1, Brx1, Ebp2, Noc2–Noc3, Nip7, Nop2 at the inter-subunit side, and Nop16 at the solvent-exposed side were not visible in the released NE1-like particles, revealing large-scale remodelling around the area that in state E is accessed and connected by the long N-terminal Erb1 tail (*Figure 4C and D*). Thus, Erb1 removal could result in destabilization of these factors, which may become invisible either by complete removal or flexible attachment to the in vitro matured pre-60S particles.

## Discussion

In this study, we provided functional insights into key restructuring events occurring on late nucleolar 60S subunit precursors preceding their transition to the nucleoplasmic compartment (*Figure 4E*). By combining biochemical and structural cryo-EM analyses, we unveiled how induction of the ATP-dependent release of the Ytm1–Erb1 heterodimer by the AAA$^+$-ATPase Rea1 results in large-scale remodelling of pre-60S particles isolated via the RNA helicase Spb4. Such in vitro matured particles, from which a network of assembly factors appeared dismantled, with subtle variations, resembled structural state NE1, while largely lacking assembly factor Nop53. NE1 is the earliest state obtained upon purification of native pre-60S particles isolated via Nop53, which can only join upon release of Erb1 due to overlapping binding sites at the pre-60S foot structure (*Kater et al., 2020*). While our biochemical data show that the removal of RNA helicase Has1 occurs together with Ytm1–Erb1, they do not suggest the complete release of further assembly factors coupled to this step. In contrast, our single-particle cryo-EM analyses clearly revealed a strong enrichment of NE1-like particle classes in the in vitro release sample, which implicates the synchronized disassembly or at least destabilization of numerous additional assembly factors. The initial release of Ytm1–Erb1 and Has1 may therefore result in pre-rRNA re-arrangement and assembly factor destabilization, which would in consequence be too flexible to be resolved by cryo-EM or become unstably attached and potentially dissociate during cryo-EM sample preparation. Indeed, this is plausible considering the intertwined nature of the Erb1 N-terminus, which, by forming various rRNA and protein contacts, may until its release stabilize nucleolar pre-60S particles. However, it remains open whether in vivo further factors and intermediary maturation steps are required to facilitate assembly factor disassembly and formation of NE1 particles upon Ytm1–Erb1 and Has1 release.

The ATP-binding pocket of Has1 was found empty in our and also a previous pre-60S cryo-EM structure (*Zhou et al., 2019b*). As furthermore a Has1 Walker A mutant protein was efficiently released from the particle in our maturation assay, we conclude that Has1 dissociates in an ATP-independent manner from 60S precursors. This is in agreement with previous studies suggesting that the Has1 function on 90S pre-ribosomes requires its catalytic activity (*Gnanasundram et al., 2019*; *Liang and Fournier, 2006*), while the role of Has1 in pre-60S maturation could be ATP-independent and may depend only on its physical presence (*Gnanasundram et al., 2019*). Alternatively, since Has1 binds already to very early nucleolar pre-60S maturation intermediates, its catalytic activity may be required during its initial pre-60S recruitment. On such early nucleolar precursors, Has1 was implicated to play a role for contact formation between the 25S 5' end and the 5.8S rRNA (*Dembowski et al., 2013*).

Our previous cryo-EM structures of nucleoplasmic pre-60S particles revealed how the pentameric Rix1-complex recruits the AAA$^+$-ATPase Rea1 to these intermediates before nuclear export (*Barrio-Garcia et al., 2016*; *Kater et al., 2020*). The Rix1-complex is thereby essential to accurately position

Rea1 for interaction and the subsequent release of its direct substrate Rsa4. Previous results (*Bassler et al., 2010*) and the increased efficiency in Ytm1–Erb1 release contributed by the Rix1-complex in our in vitro maturation assays suggest that the Rix1-complex recruits Rea1 to its earlier pre-60S substrate as well. Nevertheless, since structural insights into the more transient interaction of Rea1 with these nucleolar 60S precursors are missing, it remains unclear how the ATPase extracts Ytm1–Erb1 and how the Rix1-complex mediates this remodelling step.

The DEAD-box helicase Spb4 is one of around 20 RNA helicases involved in ribosome biogenesis that have been identified so far (*Mitterer and Pertschy, 2022*). While the approximate ribosomal maturation stages at which these RNA helicases are required have been determined, their molecular functions and precise substrates in most cases remain elusive. In addition, most RNA helicases could not be resolved on the so far existing cryo-EM structures of ribosomal precursors, likely due to high flexibility, unstable association or fast dissociation. Two exceptions in which ribosome-bound helicase structures could be determined are the DEAH helicase Dhr1 and the Ski2-like helicase Mtr4, which unwind the U3 snoRNA and RNA spacer substrates, respectively (*Cheng et al., 2020*; *Du et al., 2020*; *Lau et al., 2021*; *Schuller et al., 2018*; *Singh et al., 2021*). In contrast to such processive helicases, DEAD-box helicases are proposed to mediate RNA folding by non-processive local strand melting and annealing. In agreement with a most recently published work (*Cruz et al., 2022*), our cryo-EM structure of substrate-bound Spb4 suggests how such RNA strand displacement and re-arrangement by a DEAD-box helicase attached to its pre-ribosomal substrate is orchestrated. The existing general model on the function of DEAD-box helicases suggests that substrate RNA and ATP-binding together induce the closed active conformation of the catalytic RecA domains, whereas ATP-hydrolysis in the active site instantly triggers a re-opening of the catalytic domains going along with the release of the restructured RNA substrate.(*Andersen et al., 2006*; *Bono et al., 2006*; *Mallam et al., 2012*; *Sengoku et al., 2006*). Intriguingly, in contrast to this proposed mechanism, we found ADP instead of ATP in the nucleotide binding pocket of substrate RNA-bound Spb4 with its catalytic RecA domains in a closed conformation. Thus, while Spb4 likely has hydrolysed ATP at a previous stage, ATP-hydrolysis did not facilitate re-opening of the RecA-like domains and the helicase remained bound to ADP and its H62/H63 restructuring target. A reason for the unexpected Spb4 conformation might be the interactions of the first RecA domain with Rrp17, as well as the tight interaction of the CTD with the rRNA, which could potentially lock the rigid Spb4 core in its closed state. We therefore assume that current models on the function of DEAD-box helicases need to be extended since on complex substrates such as pre-ribosomes several additional factors might influence the state of the catalytic domains. In line with this, the DEAD-box helicase Dbp7 was also suggested to rely on an extensive interaction network for its function in rRNA folding (*Aquino et al., 2021*; *Jaafar et al., 2021*) and specific DEAD-box cofactors were shown to modulate the kinetics of the helicase core conformational cycle (*Harms et al., 2014*). Upon Spb4 dissociation, the H62/H63/H63a rRNA is then adopting a new stably folded and mature-like state as seen in cryo-EM structures of downstream pre-60S intermediates (*Kater et al., 2020*; *Wu et al., 2016*). What eventually triggers substrate release and pre-60S dissociation of Spb4 remains, however, unclear and open for future studies. As we could not recapitulate the H62/H63/H63a folding to a more mature state in our in vitro assays, the release or association of further assembly factors and ribosomal proteins may be required to facilitate this transition. While our manuscript was in revision, another study providing a highly similar cryo-EM structure of pre-60S bound Spb4 was published (*Cruz et al., 2022*). The authors report similar results on the function of Spb4 and its rRNA restructuring target and observe Spb4 in a post-catalytic state. However, they observed that the ATP-binding pocket of Spb4 is empty, whereas we clearly identified ADP in the active site. This difference may be a result of slightly different preparation conditions and indicates that after ATP hydrolysis the remaining ADP binds with low affinity.

Two recent studies reported that the human Spb4 homolog DDX55 is strongly upregulated in various lung cancers and hepatocellular carcinoma (HCC) tissues promoting HCC cell proliferation and migration (*Cui et al., 2021*; *Yu et al., 2022*). Thus, the obtained insights into the Spb4 function for ribosome biogenesis could be of relevance for medical studies and treatment of such cancer types. Furthermore, our study demonstrated the great potential of structural cryo-EM analyses of in vitro matured pre-ribosomal intermediates, which will be a promising approach to unveil further restructuring and maturation events on evolving ribosomal precursors.

## Materials and methods

### Yeast strains and plasmids

*Saccharomyces cerevisiae* strains used in this study are listed in *Supplementary file 1* and are derived from the W303 background (*Thomas and Rothstein, 1989*). Strains were constructed using established gene disruption and genomic tagging methods (*Janke et al., 2004*; *Longtine et al., 1998*). For yeast two-hybrid analyses, the reporter strain PJ69-4A was used (*James et al., 1996*). Plasmids used in this study are listed in *Supplementary file 2* and were constructed using standard DNA cloning techniques and verified by sequencing.

### Growth analyses of yeast mutant strains

To investigate growth phenotypes of plasmid-derived *spb4* mutant alleles, *LEU2* plasmids carrying the respective constructs were transformed into a *SPB4* shuffle (*spb4Δ* YCplac33-P$_{SPB4}$-*SPB4*) strain. Growth phenotypes of the mutant alleles were analysed on plates containing 1 g/l 5-fluoroorotic acid (5-FOA) (Apollo Scientific, Cat# PC4054) to select for cells that have lost the wild-type *SPB4*-containing *URA3* plasmid. To analyse growth phenotypes of plasmid-derived *rrp17* mutant alleles, a *RRP17*-HA-AID strain was transformed with plasmids carrying respective constructs and growth assessed upon Rrp17-HA-AID depletion on plates containing 2 mM auxin (3-indoleacetic acid) (Sigma-Aldrich, Cat# I2886). Genetic experiments were performed at least in two biological replicates.

### Yeast two-hybrid analyses

Plasmids expressing the Spb4 bait protein, fused to the *GAL4* DNA-binding domain (GAL4-BD), and the Rrp17 prey protein constructs, fused to the *GAL4* activation domain (GAL4-AD), were co-transformed into the reporter strain PJ69-4A (*James et al., 1996*). Yeast two-hybrid interactions were documented by spotting representative transformants in tenfold serial dilution steps on SDC-Leu-Trp and SDC-Leu-Trp-His (*HIS3* reporter gene) plates.

### Over-expression and affinity purification of yeast proteins

An N-terminal protein A-TEV-Cbp-HA (TAP-HA)-*REA1* fusion was over-expressed from a centromeric plasmid under the control of the galactose-inducible *GAL1-10* promoter. Cells were grown in synthetic galactose complete medium lacking leucine (SGC-Leu) to an OD$_{600}$ value of around 1.5. The medium was supplemented with 1.2% galactose, 1.2% bacto peptone, and 0.6% yeast extract (final concentrations) and cells further grown to an OD$_{600}$ value of ~6. Cells were harvested and flash-frozen in liquid nitrogen and stored at –20°C. Cells were lysed in lysis buffer containing 25 mM HEPES (pH 7.5), 250 mM NaCl, 10 mM MgCl$_2$, 10 mM KCl, 5% glycerol, 0.01% NP-40, 1 mM DTT, 0.5 mM phenylmethylsulfonyl fluoride (PMSF), and 1× SIGMAFAST protease inhibitor (Sigma-Aldrich, Cat# S8830), by shaking in a bead beater (Fritsch) in the presence of glass beads. The lysate was cleared by consecutive centrifugation at 5000 and 18,000 rpm and incubated with IgG Sepharose 6 Fast Flow beads (GE Healthcare; Cat# 17096902) on a rotating wheel for 120 min at 4°C. Beads were extensively washed with lysis buffer (without protease inhibitors) and subsequently eluted through TEV cleavage at 16°C for 150 min. A final concentration of 2 mM CaCl$_2$ was added to the protein eluate prior to incubation with Calmodulin agarose beads (G-Biosciences; Cat# 786–282) for 90 min at 4°C. After extensive washing with lysis buffer containing 2 mM CaCl$_2$, the protein was eluted by three consecutive incubations with elution buffer containing 50 mM TRIS (pH 8.0), 100 mM NaCl, 5 mM KCl, 2 mM MgCl$_2$, 8 mM EGTA, 0,01% NP-40 for 20 min at 4°C. EGTA-eluates were pooled and concentrated to a final concentration of ~2 mg/ml using an Amicon Ultra-4 cellulose centrifugal filter unit (Merck Millipore, Cat# UFC810024).

The pentameric Rix1$_{(2)}$–Ipi3$_{(2)}$–Ip1 complex, was over-expressed from 2 micron plasmids harbouring *RIX1*-TEV-proteinA, *IPI1*-Cbp, and untagged *IPI3* under control of the inducible *GAL1-10* promoter. Cells were grown in synthetic raffinose complete medium lacking leucine and tryptophane (SRC-Leu-Trp) to an OD$_{600}$ value of around 1.5. Protein co-expression was induced by addition of an equal volume of double concentrated YPG medium for 6–7 hr. Cell harvesting and protein purification until elution with EGTA (in buffer without NP-40) was performed as described above. EGTA-eluates were concentrated and further purified by size exclusion chromatography (SEC) on a Superose 6 increase 10/300 GL column (GE Healthcare) equilibrated with lysis buffer. The respective fractions were pooled and concentrated to a final concentration of ~2 mg/ml using Amicon Ultra-4 cellulose filter units.

## Affinity purification of pre-ribosomal 60S particles

For two-step affinity purifications from yeast, respective Flag-TEV-proteinA (FTpA) tagged bait proteins were expressed under control of their native promoter. Yeast strains were grown in YPD (for endogenous bait proteins) or in synthetic dextrose complete (SDC) medium lacking leucine (for plasmid-derived bait proteins or maintenance of plasmid-derived mutants) at 30°C. Cells were harvested in the logarithmic growth phase, flash frozen in liquid nitrogen, and stored at –20°C. For depletion of auxin-inducible degron (AID) tagged proteins (as indicated in the figures), cultures were incubated in the presence of 0.5 mM auxin (3-indoleacetic acid, Sigma-Aldrich, Cat# I2886) for 120 min prior to harvesting the cells. Cell pellets were resuspended in lysis buffer containing 50 mM Tris-HCl (pH 7.5), 100 mM NaCl, 5 mM $MgCl_2$, 0.05% NP-40, 1 mM DTT, supplemented with 1 mM PMSF, 1 × SIGMAFAST protease inhibitor (Sigma-Aldrich), and cells were ruptured by shaking in a bead beater (Fritsch) in the presence of glass beads. Lysates were cleared by two subsequent centrifugation steps at 4°C for 10 and 30 min at 5000 and 15,000 rpm, respectively. Supernatants were incubated with immunoglobulin G (IgG) Sepharose 6 Fast Flow beads (GE Healthcare) on a rotating wheel at 4°C for 90 min. Beads were transferred into Mobicol columns (MoBiTec) and, after washing with 15 ml of lysis buffer (per 2 l culture volume), cleavage with tobacco etch virus (TEV) protease was performed at 16°C for 120 min. In a second purification step, TEV eluates were incubated with Flag agarose beads (ANTI-FlagM2 Affinity Gel, Sigma-Aldrich, Cat# A2220) for 75 min at 4°C. After washing with 5 ml of lysis buffer, bound proteins were eluted with lysis buffer containing 300 µg/ml Flag peptide (Sigma-Aldrich, Cat# F3290) at 4°C for 45 min. Buffer lacking NP-40 was used for the last purification step in samples used for cryo-EM. Flag eluates were analysed by SDS-PAGE on 4–12% polyacrylamide gels (NuPAGE, Invitrogen, Cat# NP0322BOX) with colloidal Coomassie staining (Roti-blue, Carl Roth, Cat# A152.1) or by western blotting with antibodies, as indicated in the respective figures. Pre-60S affinity purifications were performed at least in two technical replicates.

## In vitro maturation assays with purified pre-ribosomal 60S particles

Yeast strains expressing chromosomal *SPB4*-FTpA (or plasmid derived YCplac111-*SPB4*-FTpA and YCplac111-*spb4*_R360A-FTpA for experiments in *Figure 4—figure supplement 1B*), *RRP17*-FTpA, or TAP-HA-*YTM1 RPL3*-Flag fusions under control of the respective endogenous promoters were grown in YPD or SDC-Leu medium and as indicated in the figures incubated in the presence of 0.5 mM auxin (3-indoleacetic acid, Sigma-Aldrich) for 120 min prior to harvesting the cells. Pre-ribosomal particles were purified via immunoglobulin G (IgG) Sepharose 6 Fast Flow beads (GE Healthcare) as described above. TEV-eluates were adjusted to a final volume of 0.5 ml and, as indicated in the figures, incubated with ATP (3 mM final concentration) (Sigma-Aldrich, Cat# A2383), purified Rea1 protein, and/or purified Rix1-complex on a rotating wheel for 45 min at room temperature. Subsequently, samples were chilled on ice and still bound material was re-isolated by incubation with anti-Flag agarose beads (ANTI-FlagM2 Affinity Gel, Sigma-Aldrich) for 60 min at 4°C and eluted and analysed as described above. The release assays with Spb4 particles were performed at least in three technical replicates.

## Western blotting

Western blot analysis was performed using the following antibodies: anti-Nog1 antibody (1:5000), anti-Nog2 antibody (1:20,000), anti-Arx1 antibody (1:2,000), anti-Nsa2 antibody (1:10,000), anti-Rlp24 antibody (1:2,000), provided by Micheline Fromont-Racine, anti-Nug1 antibody (1:10,000), anti-Bud20 antibody (1:5000), provided by Vikram Panse, anti-Ytm1 antibody (1:100), provided by John Woolford, anti-Has1 antibody (1:10,000) provided by Patrick Linder, anti-Ebp2 antibody (1:10,000), provided by Keiko Mizuta, anti-Noc3 antibody (1:1000), provided by Herbert Tschochner, anti-Rpl3 antibody (1:5,000), provided by Jonathan Warner, anti-Rsa4 antibody (1:10,000), provided by Miguel Remacha, anti-Arc1 antibody (1:5000, raised in the Hurt lab), horseradish-peroxidase-conjugated anti-Flag antibody (1:10,000; Sigma-Aldrich, Cat# A8592, RRID:AB_439702), horseradish-peroxidase-conjugated anti-HA antibody (1:5000; Roche, Cat# 12013819001; RRID:AB_390917), secondary horseradish-peroxidase-conjugated goat anti-rabbit antibody (1:2,000; Bio-Rad, Cat# 166-2408EDU, RRID:AB_11125345), and secondary horseradish-peroxidase-conjugated goat anti-mouse antibody (1:2,000; Bio-Rad, Cat# STAR105P, RRID:AB_323002).

## Cryo-EM sample preparation, analysis, and image processing

All purified samples were supplemented with 0.05% octaethylene glycol monododecyl ether. The samples (3.5 µl) were applied onto Quantifoil R3/3 Holey grids with 2 nm continuous carbon support, which were glow discharged with a Harrick PDC-32G plasma cleaner at $2.1 \times 10^{-1}$ torr for 20 s. After a 45 s incubation time, the grids were blotted for 3 s and plunge frozen in liquid ethane using a Vitrobot Mark IV (FEI Company) operated at 4°C and 90% humidity.

Cryo-EM data were collected on a Titan Krios operated at 300 kV, equipped with a Gatan K2 direct electron detector. Images were acquired with a nominal pixel size of 1.059 Å under low-dose conditions, with a total dose of 42.4–46.8 e⁻/Å² over 40 frames using EPU (Thermo Fisher). Gain corrected frames were dose-weighted, aligned, summed, and motion corrected with MotionCor2 (*Zheng et al., 2017*) and the contrast-transfer-function (CTF) parameters were estimated with CTFFIND4 and Gctf (*Rohou and Grigorieff, 2015*; *Zhang, 2016*).

The micrographs were manually inspected and low-quality micrographs were discarded. Particles were auto-picked without reference in Relion 3.1 (*Zivanov et al., 2020*; *Zivanov et al., 2018*) using the Laplacian-of-Gaussian blob detection. The particles were extracted and imported into CryoSPARC (*Punjani et al., 2017*) for iterative 2D classification and good 2D classes were selected and used for homogeneous refinement with the EMD-3891 map (lowpass filtered to 60 Å) as an initial reference (*Kater et al., 2017*). Particles were reimported into Relion 3.1 using the pyem (csparc2star.py) tool and 3× binned particles were extracted with 3.177 Å/pixel. All subsequent 3D refinements and 3D classifications steps were performed with Relion 3.1. For details see sorting schemes in *Figure 3—figure supplement 1*, *Figure 4—figure supplement 3*, and *Figure 4—figure supplement 3*. The particles of the final classes were extracted with a pixel size of 1.059 Å/pixel, and homogeneous refinements, local refinements, and local filtering were performed with CryoSPARC. Composite maps were generated with ChimeraX using the vop max command (*Goddard et al., 2018*; *Pettersen et al., 2021*).

For the uniform data processing and comparison of the untreated Spb4 control sample and the in vitro release sample, the micrographs of both dataset were joined in Relion 3.1 and processed as described above. Particle numbers of the Spb4 control dataset and the in vitro release dataset of each class were obtained through the unique optic groups identifiers of the datasets.

## Model building and refinement

The model PDB-6ELZ (*Kater et al., 2017*) was fitted into the density map of the State E particle and was used as an initial model. The overall better resolution of our cryo-EM map compared to the original reconstruction allowed us to improve the state E model (PDB-6ELZ). In particular, molecular models of Rrp17, Loc1, Noc2, and parts of Spb1 were built de novo in Coot (*Emsley et al., 2010*; *Emsley and Cowtan, 2004*). For the Spb4 model, swiss-model (*Waterhouse et al., 2018*) and alphaFold models (*Jumper et al., 2021*) were used as initial references. The RecA domains were rigid body fitted separately into the density and the C-terminus was built de novo. For the state D, we used our molecular model of the state E, the PDB entries 6C0F (*Sanghai et al., 2018*) and 6EM5 (*Kater et al., 2017*) as initial references. The model of state E without Rrp17 was generated with our model of state E as initial reference. The model was rigid body fitted into the cryo-EM map and badly resolved and/or missing parts including Rrp17 and the Spb1-MTD were deleted. For the release state particle, the state E model and the NE1 state model PDB-6YLX (*Kater et al., 2020*) were used as reference. All models were manually adjusted in Coot (*Emsley et al., 2010*; *Emsley and Cowtan, 2004*) and the final models were real-space refined using Phenix (*Adams et al., 2010*; *Liebschner et al., 2019*). Data collection and refinement statistics are summarized in *Supplementary file 3*.

ChimeraX was used to visualize molecular models and cryo-EM densities (*Goddard et al., 2018*; *Pettersen et al., 2021*). The model-based secondary structure predictions of rRNA H62/H63/H63a and the ITS2 were generated with Forna (*Kerpedjiev et al., 2015*).

## Acknowledgements

We thank C Ungewickell and S Rieder for cryo-EM data collection. We thank Dieter Kressler for providing the *SPB4* shuffle strain and Petra Ihrig and Jürgen Reichert from the BZH Heidelberg MS facility for performing mass spectrometry analyses.

This work was supported by grants from the German Research Council (HU363/15-2 to EH), and from the European Research Council (885711-Human-Ribogenesis to RB and ADG 741781 GLOWSOME to EH).

## Additional information

### Funding

| Funder | Grant reference number | Author |
|---|---|---|
| European Research Council | 885711-Human-Ribogenesis | Roland Beckmann |
| Deutsche Forschungsgemeinschaft | HU363/15-2 | Ed Hurt |
| European Research Council | ADG 741781 GLOWSOME | Ed Hurt |

The funders had no role in study design, data collection and interpretation, or the decision to submit the work for publication.

### Author contributions

Valentin Mitterer, Conceptualization, Resources, Data curation, Formal analysis, Validation, Investigation, Visualization, Methodology, Writing – original draft, Project administration, Writing – review and editing; Matthias Thoms, Conceptualization, Resources, Data curation, Software, Formal analysis, Validation, Investigation, Visualization, Methodology, Project administration, Writing – review and editing; Robert Buschauer, Software, Formal analysis; Otto Berninghausen, Data curation; Ed Hurt, Roland Beckmann, Funding acquisition, Project administration, Writing – review and editing

### Author ORCIDs

Valentin Mitterer ⬤ http://orcid.org/0000-0003-1587-1194
Matthias Thoms ⬤ http://orcid.org/0000-0001-8084-6097
Roland Beckmann ⬤ http://orcid.org/0000-0003-4291-3898

### Decision letter and Author response

Decision letter https://doi.org/10.7554/eLife.84877.sa1
Author response https://doi.org/10.7554/eLife.84877.sa2

## Additional files

### Supplementary files

- Supplementary file 1. *Saccharomyces cerevisiae* strains used in this study.
- Supplementary file 2. Plasmids used in this study.
- Supplementary file 3. Data collection and refinement statistics.
- MDAR checklist

### Data availability

Atomic models reported in this study have been deposited in the Protein Data Bank (PDB) and can be retrieved using the following accession codes: 8BVN, 8BVU, 8BVV, 8BVY. Cryo-EM density maps have been deposited in the Electron Microscopy Data Bank (EMDB) and can be retrieved using the following accession codes: 16267, 16272, 16273, 16275, 16276, 16277, 16278. Yeast strains and plasmids are available from the corresponding authors upon request.

The following datasets were generated:

| Author(s) | Year | Dataset title | Dataset URL | Database and Identifier |
|---|---|---|---|---|
| Thoms M, Mitterer V, Buschauer R, Berninghausen O, Hurt E, Beckmann R | 2022 | State E pre-60S subunit | https://www.rcsb.org/structure/unreleased/8BVN | RCSB Protein Data Bank, 8BVN |
| Thoms M, Mitterer V, Buschauer R, Berninghausen O, Hurt E, Beckmann R | 2023 | State D pre-60S subunit | https://www.rcsb.org/structure/unreleased/8BVU | RCSB Protein Data Bank, 8BVU |
| Thoms M, Mitterer V, Buschauer R, Berninghausen O, Hurt E, Beckmann R | 2023 | StateE without Rrp17, pre-60S subunit | https://www.rcsb.org/structure/unreleased/8BVV | RCSB Protein Data Bank, 8BVV |
| Thoms M, Mitterer V, Buschauer R, Berninghausen O, Hurt E, Beckmann R | 2022 | Release state pre-60S subunit | https://www.rcsb.org/structure/unreleased/8BVY | RCSB Protein Data Bank, 8BVY |
| Thoms M, Mitterer V, Buschauer R, Berninghausen O, Hurt E, Beckmann R | 2022 | State E pre-60S | http://www.ebi.ac.uk/pdbe/entry/emdb/EMD-16267 | Electron Microscopy Data Bank, EMD-16267 |
| Thoms M, Mitterer V, Buschauer R, Berninghausen O, Hurt E, Beckmann R | 2022 | State D pre-60S | https://www.ebi.ac.uk/emdb/EMD-16272 | Electron Microscopy Data Bank, EMD-16272 |
| Thoms M, Mitterer V, Buschauer R, Berninghausen O, Hurt E, Beckmann R | 2022 | State E without Rrp17 | http://www.ebi.ac.uk/pdbe/entry/emdb/EMD-16273 | Electron Microscopy Data Bank, EMD-16273 |
| Thoms M, Mitterer V, Buschauer R, Berninghausen O, Hurt E, Beckmann R | 2022 | Release state | http://www.ebi.ac.uk/pdbe/entry/emdb/EMD-16275 | Electron Microscopy Data Bank, EMD-16275 |
| Thoms M, Mitterer V, Buschauer R, Berninghausen O, Hurt E, Beckmann R | 2022 | Release state ES27up | http://www.ebi.ac.uk/pdbe/entry/emdb/EMD-16276 | Electron Microscopy Data Bank, EMD-16276 |
| Thoms M, Mitterer V, Buschauer R, Berninghausen O, Hurt E, Beckmann R | 2022 | Release state with Spb4 | http://www.ebi.ac.uk/pdbe/entry/emdb/EMD-16277 | Electron Microscopy Data Bank, EMD-16277 |
| Thoms M, Mitterer V, Buschauer R, Berninghausen O, Hurt E, Beckmann R | 2022 | Release state without Nop53 | http://www.ebi.ac.uk/pdbe/entry/emdb/EMD-16278 | Electron Microscopy Data Bank, EMD-16278 |

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
