## [Editor Report]

This fundamental study substantially advances our understanding of the process of ribosome maturation. The authors have purified and determined the structures of several nucleolar ribosome assembly intermediates in yeast using cryo-electron microscopy (cryo-EM). The study combines genetic, biochemical, and structural analysis to provide compelling support for the conclusions the authors wish to draw. The work will be of broad interest to cell biologists, biochemists, and structural biologists.

---

## [Decision Letter]

**Decision letter after peer review:**

Thank you for submitting your article "Concurrent remodelling of nucleolar 60S subunit precursors by the Rea1 ATPase and Spb4 RNA helicase" for consideration by *eLife*. Your article has been reviewed by 3 peer reviewers, one of whom is a member of our Board of Reviewing Editors, and the evaluation has been overseen by Ruben Gonzalez as the Reviewing Editor and James Manley as the Senior Editor. The following individual involved in the review of your submission has agreed to reveal their identity: Arlen W Johnson (Reviewer #3).

Essential revisions:

1) As requested by all three reviewers, the authors must revise the text to acknowledge, cite, and discuss the recent publication by Cruz et al., (https://doi.org/10.1038/s41594-022-00874-9) in all the appropriate places in the manuscript.

2) The authors should address the remaining points listed in Reviewer 2's Comments for the Authors and the Major Points raised in Reviewer 3's Public Review and Comments for the Authors.

*Reviewer #1 (Recommendations for the authors):*

The observation that the RecA domains of Spb4 are in their closed state while Spb4 is bound to ADP in the cryo-EM structure of the Spb4 particle as well as the interpretation and mechanistic implications of this observation is a major finding of this work. Nonetheless, I wonder whether the resolution of the structure truly allows the authors to distinguish between ADP and ATP. The segmented density for ADP shown in the model of the nucleotide-binding site of Spb4 in Figure 3M doesn't really allow an assessment of this issue. Given the importance of this observation to the work, the authors should provide a deeper assessment of the data in this regard and present a more complete argument excluding the possibility that ATP (or ADP-Pi, see below) might occupy the nucleotide-binding site. More importantly, the authors' conclusion in this regard is supported by the recent publication of Cruz et al., (https://doi.org/10.1038/s41594-022-00874-9), and this publication should be discussed and cited in this context.

Related to the previous point, given that the RecA domains are closed, even if ATP has been hydrolyzed, I wonder whether and how the inorganic phosphate (Pi) has dissociated from the nucleotide-binding site. In my view, the question of whether and how the closing/opening dynamics of the RecA domains of DEAD-box helicases are coupled to ATP binding, ATP hydrolysis, and/or π release field remains open in the field. My understanding is that the majority of the field expects closing to be coupled to ATP binding and opening to be coupled to ATP hydrolysis or π release. Interestingly, previous work from the Klostermeier group on a different DEAD-box helicase, eIF4A, suggests that the closing/opening dynamics are faster than ATP binding, ATP hydrolysis, or π release (Harms, et al. (2014) NAR 42, 7911), but the mechanism ultimately proposed in that work is that closing is coupled to ATP binding and opening is coupled to π release, in line with what I think is the predominant view. In this context, I think the findings reported here will be of great interest and importance to the DEAD-box helicase field. Assuming they can strengthen their conclusion that Spb4 is bound to ADP (see previous point), the authors should expand their discussion of these issues and contextualize their findings that the RecA domains of Spb4 are closed while Spb4 is bound to ADP relative to previous work and mechanistic models in the field. Related to this, I agree with the authors that the contacts the C-terminal domain of Spb4 makes with the substrate might be relevant and might uniquely stabilize Spb4 in a conformation in which the RecA domains are in the closed conformation while Spb4 is bound to ADP.

*Reviewer #2 (Recommendations for the authors):*

1. "Our cryo-EM structure of substrate-bound Spb4 for the first time suggests how such RNA strand displacement and re-arrangement by a DEAD-box helicase attached to its pre-ribosomal substrate is orchestrated". Please modify the text and incorporate some discussion of the recently published work on Spb4 (Cruz et al., https://doi.org/10.1038/s41594-022-00874-9) into the manuscript.

2. P7 line 221: The authors discuss the identification of the likely SAH ligand and the 2'-O-ribose methylation of G2922. More context should be provided on the functional role of this modification in light of the recent study from the Taylor/Johnson labs which should be cited in the current manuscript (Yelland, J.N., Bravo, J.P.K., Black, J.J. et al. A single 2′-O-methylation of ribosomal RNA gates assembly of a functional ribosome. Nat Struct Mol Biol (2022). https://doi.org/10.1038/s41594-022-00891-8).

3. On p9 line 266, the authors suggest that Spb4 "induces" bending and strand separation of the rRNA at the base of ES27. They also suggest that the CTD of Spb4 "induces" substrate RNA strand disruption. However, no experimental evidence is provided to support these claims. The authors should consider citing the global SHAPE analysis showing that the H62-H63-ES27 region forms stable duplexes (Burlacu et al. 2017) and footprinting experiments showing that Spb4 disrupts this base pairing (Bruning et al. 2018) (also further analysed by Cruz et al. 2022) as this seems to provide further support their proposed model.

4. Can the authors comment more broadly on what insights their maps provide on the formation of the H61-H64 junction and the H61-H62-H64 three-way junction?

5. "Furthermore, a conserved tryptophan (W536) within the flexible C-terminal tail of the CTD (aa 500-606) intercalates between nucleotides of helices H62/H63/H63a and thereby might prevent this rRNA region to adopt its mature-like fold as seen in the nucleoplasmic pre-60S particles (Figures 3C and 3L)". This seems like a reasonable hypothesis, but are there any experimental data to support it? Please comment.

6. The authors should reference the recent publications from the Bohnsack and Henras groups showing how the related RNA DEAD-box helicase Dbp7 functions in pre-rRNA folding.

7. P4, line 109: the authors write "our high-resolution cryo-EM structure". What does this term mean? It would be preferable to remove this term and simply state a range of resolutions.

8. Figure 1- Please clarify in the legend what II, II, and VI signify?

9. Figure 4A-The presentation of the input +/- at the top of the panel is missing some labels. Please correct.

10. Some of the details in the figures are difficult to see in the pdf file eg Figure 3L, the Trp residue is unclear. Please try to improve the presentation/labelling.

---

## [Author Response]

Reviewer #1 (Recommendations for the authors):The observation that the RecA domains of Spb4 are in their closed state while Spb4 is bound to ADP in the cryo-EM structure of the Spb4 particle as well as the interpretation and mechanistic implications of this observation is a major finding of this work. Nonetheless, I wonder whether the resolution of the structure truly allows the authors to distinguish between ADP and ATP. The segmented density for ADP shown in the model of the nucleotide-binding site of Spb4 in Figure 3M doesn't really allow an assessment of this issue. Given the importance of this observation to the work, the authors should provide a deeper assessment of the data in this regard and present a more complete argument excluding the possibility that ATP (or ADP-Pi, see below) might occupy the nucleotide-binding site. More importantly, the authors' conclusion in this regard is supported by the recent publication of Cruz et al., (https://doi.org/10.1038/s41594-022-00874-9), and this publication should be discussed and cited in this context.

We agree that the Figure 3M doesn’t allow for a sufficient assessment of the mentioned nucleotide issue. We therefore changed main Figure 3M and added another figure to the supplement (Figure3—figure supplement 4E) displaying this region of the map in a clearer way and together with rigid body fitted RNA helicase models containing ADP, the ATP analogue ADPNP, ADP BeF3 or ADP PI. It can be better appreciated now that the map corresponds to an ADP molecule without any additional density to accommodate the γ-phosphate of an ATP molecule. The local resolution of this region of about 3.0 to 3.5 Å is sufficient to draw this conclusion unambiguously. This notion is further supported by the finding that we can observe density for a nearby magnesium ion in its expected position.

As suggested by the reviewer, we discuss this issue in the context of the supporting findings or Cruz et al. in the revised text.

Related to the previous point, given that the RecA domains are closed, even if ATP has been hydrolyzed, I wonder whether and how the inorganic phosphate (Pi) has dissociated from the nucleotide-binding site. In my view, the question of whether and how the closing/opening dynamics of the RecA domains of DEAD-box helicases are coupled to ATP binding, ATP hydrolysis, and/or π release field remains open in the field. My understanding is that the majority of the field expects closing to be coupled to ATP binding and opening to be coupled to ATP hydrolysis or π release. Interestingly, previous work from the Klostermeier group on a different DEAD-box helicase, eIF4A, suggests that the closing/opening dynamics are faster than ATP binding, ATP hydrolysis, or π release (Harms, et al. (2014) NAR 42, 7911), but the mechanism ultimately proposed in that work is that closing is coupled to ATP binding and opening is coupled to π release, in line with what I think is the predominant view. In this context, I think the findings reported here will be of great interest and importance to the DEAD-box helicase field. Assuming they can strengthen their conclusion that Spb4 is bound to ADP (see previous point), the authors should expand their discussion of these issues and contextualize their findings that the RecA domains of Spb4 are closed while Spb4 is bound to ADP relative to previous work and mechanistic models in the field. Related to this, I agree with the authors that the contacts the C-terminal domain of Spb4 makes with the substrate might be relevant and might uniquely stabilize Spb4 in a conformation in which the RecA domains are in the closed conformation while Spb4 is bound to ADP.

We have added some discussion in which we better explain the general model on closure/opening of the catalytic RecA domains. Furthermore, we proposed in the discussion that on complex substrates such as pre-ribosomes the states of the domains may be regulated in a more complicated manner than so far expected and referred also to the work of the Klostermeier group (Harms et al. 2014) revealing that additional factors/cofactors can modulate the conformation of the catalytic domain.

Reviewer #2 (Recommendations for the authors):1. "Our cryo-EM structure of substrate-bound Spb4 for the first time suggests how such RNA strand displacement and re-arrangement by a DEAD-box helicase attached to its pre-ribosomal substrate is orchestrated". Please modify the text and incorporate some discussion of the recently published work on Spb4 (Cruz et al., https://doi.org/10.1038/s41594-022-00874-9) into the manuscript.

We have modified the sentence accordingly and added a citation of the study recently published by Cruz et al. Furthermore, we are citing the study in the introduction and discussion, and we are pointing out the major difference between the two structures regarding the nucleotide binding state of Spb4.

2. P7 line 221: The authors discuss the identification of the likely SAH ligand and the 2'-O-ribose methylation of G2922. More context should be provided on the functional role of this modification in light of the recent study from the Taylor/Johnson labs which should be cited in the current manuscript (Yelland, J.N., Bravo, J.P.K., Black, J.J. et al. A single 2′-O-methylation of ribosomal RNA gates assembly of a functional ribosome. Nat Struct Mol Biol (2022). https://doi.org/10.1038/s41594-022-00891-8).

We added a sentence on the importance of the G2922 methylation that is recognized by Nog2 during pre-60S maturation, which was recently revealed by the mentioned work of the Johnson/Taylor labs. As suggested by Reviewer #3, we furthermore cited the publication by Lapeyre and Purushothaman in this context.

3. On p9 line 266, the authors suggest that Spb4 "induces" bending and strand separation of the rRNA at the base of ES27. They also suggest that the CTD of Spb4 "induces" substrate RNA strand disruption. However, no experimental evidence is provided to support these claims. The authors should consider citing the global SHAPE analysis showing that the H62-H63-ES27 region forms stable duplexes (Burlacu et al. 2017) and footprinting experiments showing that Spb4 disrupts this base pairing (Bruning et al. 2018) (also further analysed by Cruz et al. 2022) as this seems to provide further support their proposed model.

We added a sentence pointing out that the proposed induced bending and disruption of the H62/H63/H63a rRNA by Spb4 is in agreement with existing structural probing data.

4. Can the authors comment more broadly on what insights their maps provide on the formation of the H61-H64 junction and the H61-H62-H64 three-way junction?

With respect to junction formation, the overall positioning of rRNA H61-H64 does not change substantially between our Spb4-bound state E and the known subsequent states after Spb4 release, and also the junction directly connecting H61-H64 is already established and doesn’t change. The H61-H62-H64 three-way junction, however, is not well resolved in our Spb4-bound and also not in the Spb4 post-release maps. Therefore, our maps do not provide precise insights on the exact conditions (binding of additional AFs or ribosomal proteins might be required) and time point for the maturation of this junction, apart from the observation that in the Spb4-bound state it is partially disordered and adopts in the known later states (NE1) its defined mature conformation.

5. "Furthermore, a conserved tryptophan (W536) within the flexible C-terminal tail of the CTD (aa 500-606) intercalates between nucleotides of helices H62/H63/H63a and thereby might prevent this rRNA region to adopt its mature-like fold as seen in the nucleoplasmic pre-60S particles (Figures 3C and 3L)". This seems like a reasonable hypothesis, but are there any experimental data to support it? Please comment.

As we have no experimental data to support this hypothesis, we adjusted this sentence accordingly.

6. The authors should reference the recent publications from the Bohnsack and Henras groups showing how the related RNA DEAD-box helicase Dbp7 functions in pre-rRNA folding.

We are now citing these two publications in the discussion.

7. P4, line 109: the authors write "our high-resolution cryo-EM structure". What does this term mean? It would be preferable to remove this term and simply state a range of resolutions.

We have removed the unspecific term “high-resolution”.

8. Figure 1- Please clarify in the legend what II, II, and VI signify?

The numbers signify the catalytic helicase core signature motifs I, II, and VI. We have now clarified this in the figure legend.

9. Figure 4A-The presentation of the input +/- at the top of the panel is missing some labels. Please correct.

Thank you for pointing this out, we have added the missing labels in Figure 4A.

10. Some of the details in the figures are difficult to see in the pdf file eg Figure 3L, the Trp residue is unclear. Please try to improve the presentation/labelling.

As suggested, we improved the presentation and labelling in Figure 3L.